# Efficient Compression of Time-Series Foundation Models via Consensus Subspace Distillation

## Abstract

Compressing universal time-series foundation models (TSFMs) significantly reduces computational and storage overhead, thereby facilitating their widespread adoption. In TSFM compression techniques, knowledge distillation stands out by transferring knowledge from teacher models to student models. However, existing distillation methods often overlook the inherent consensus representation spaces in TSFMs and the imbalance in hierarchical contributions, leading to inefficient knowledge transfer. To address this, we propose a novel approach that reformulates distillation as a consensus subspace optimization task, leveraging the observation that high-level embeddings autonomously converge across different model scales, along with the long-tail distribution of hierarchical contributions. We tackle the consensus subspace problem by identifying and extracting scale-invariant low-rank subspaces: on local data subsets, we perform singular value decomposition on embeddings from offline-selected consensus layers to derive consensus projection matrices, which are then used to fine-tune the student model, ensuring representation alignment and accelerated convergence. Additionally, we introduce a scalable uncertainty injection mechanism to enhance generalization to unseen data, modeling subset biases as frequency-domain gaps to inflate covariances. Extensive experiments demonstrate that our framework excels on multiple standard time-series datasets, with student models even surpassing teacher performance in time-series forecasting tasks. Compared to state-of-the-art methods, our approach achieves over 90% parameter reduction and 100x distillation speedup while retaining comparable performance across various time-series tasks. Code and compressed model weights are available via an anonymous link: `anonymous.4open.science/r/CSD-13C3/`.

## 1 Introduction

Transformer-based time series foundation models (TSFMs) have significantly advanced the processing of complex sequential data. These models enable multitask generalization and robust predictions across various domains (Liang et al., 2024). However, as model scales grow, the associated computational and storage overheads rise substantially. This limits their deployment in resource-constrained environments. To address this challenge, model compression techniques have become essential. They compress large TSFMs into efficient versions while preserving performance as much as possible (Liu et al., 2025; Shi et al., 2025).

Among compression strategies for TSFMs, several approaches stand out for reducing model size and inference costs. These include neural architecture search (NAS), pruning, knowledge distillation (KD), quantization, and low-rank mapping (Fournier et al., 2023). NAS automatically designs efficient architectures, though it often involves high search costs (Wang et al., 2024). Pruning simplifies models by removing redundant weights, but it may impair the representational capacity of critical hierarchical structures (Xu et al., 2022). Quantization reduces numerical precision for compression, yet improper tuning can compromise generalization on long sequences (Li et al., 2024). Low-rank mapping captures key information subspaces, but it often overlooks hierarchical imbalances in Transformers (Zha et al., 2024). In contrast, knowledge distillation transfers knowledge from teacher models to student models effectively. It maintains consensus representations and addresses

hierarchical imbalances in TSFMs. This makes it a promising choice for efficient knowledge transfer without training from scratch.

A core challenge in distilling TSFMs lies in efficiently transferring hierarchical knowledge while preserving the model's inherent characteristics. Traditional distillation methods fall into three categories: response-based, feature-based, and relation-based. Response-based KD aligns output logits or soft labels to emphasize probabilistic imitation (Hinton et al., 2015). However, it often overlooks dynamic information in intermediate layers (see Fig. 1(a)). Feature-based KD matches intermediate activations, but this rigid binding disrupts natural convergence to scale-invariant subspaces and amplifies low-level noise (Fig. 1(b)) (Romero et al., 2014; Zhu & Zhang, 2025). Relation-based KD focuses on inter-sample or inter-layer similarities, such as attention maps (Park et al., 2019a). Yet it neglects the self-organizing alignment of high-level embeddings across model scales. This leads to low distillation efficiency and insufficient generalization, especially when layer contributions follow long-tail distributions (Fig. 1(c)).

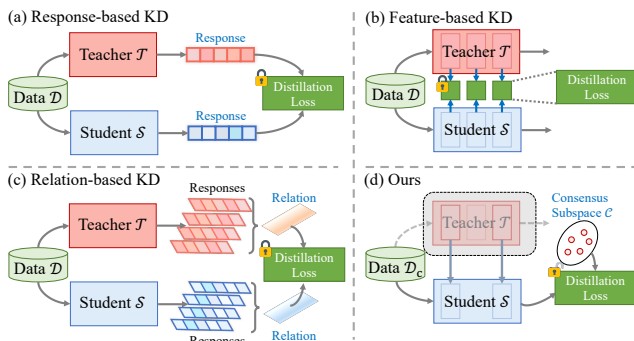

Figure 1: Comparison of four knowledge distillation (KD) paradigms. (a) Response-based distillation aligns only output probabilities and ignores dynamic information in intermediate layers. (b) Feature-based distillation enforces activation matching but disrupts natural convergence processes. (c) Relation-based distillation focuses on inter-sample similarity yet overlooks cross-layer self-organizing consistency. (d) Our proposed consensus subspace optimization method extracts scale-invariant low-rank subspaces to guide student models toward geometric centers. This avoids dependence on teacher-specific pathways.

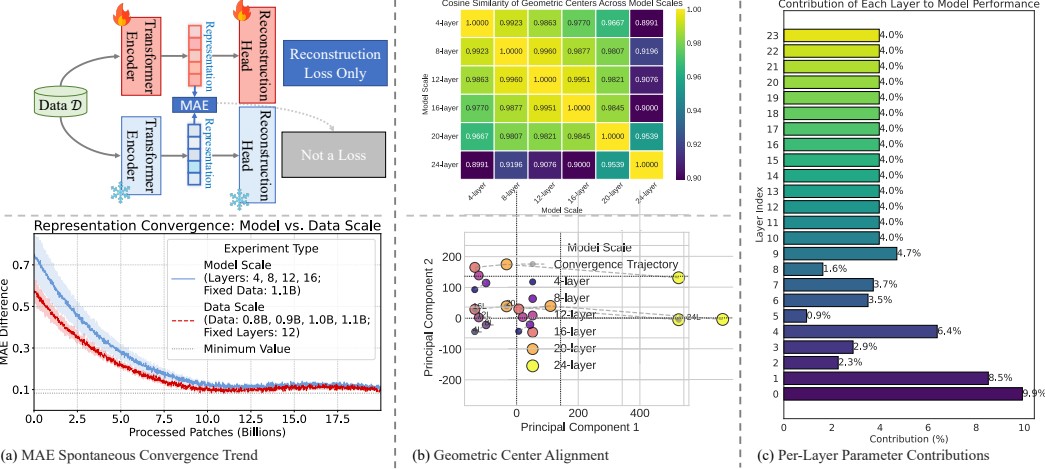

(a) MAE Spontaneous Convergence Trend

(b) Geometric Center Alignment

(c) Per-Layer Parameter Contributions

Figure 2: Empirical motivation validation. (a) During pre-training of time series foundation models across varying data and model scales, representations spontaneously converge to align with those of larger pre-trained models under unconstrained conditions. The upper panel illustrates the convergence measurement methodology, while the lower panel shows results with error bands across scales. (b) Projected representations reveal geometrically aligned centers across scales. The upper panel displays a cosine similarity heatmap of geometric centers, indicating highly aligned spaces. The lower panel depicts centroid offset trajectories in a reference frame (using the 24-layer model's center as origin), demonstrating tight clustering and convergence via PCA projection. Together, (a) and (b) suggest a potential consensus subspace with an invariant geometric center. (c) Bar chart of per-layer parameter contributions, showing a long-tail distribution.

These limitations become evident in empirical studies. During masked autoencoder (MAE) pretraining on time series, high-level embeddings from models of varying scales, such as 12-layer and 24-layer, tend to converge to consistent consensus subspaces. They show minimal unconstrained MAE differences (see Fig. 2(a)) and highly aligned projection centers (Fig. 2(b)). However, existing methods fail to leverage this phenomenon. Instead, they bind student models to teacher-specific paths and overlook the independent contributions of shallow layers to context capture (Fig. 2(c)). This results in amplified biases and slower convergence.

To overcome these issues, we propose a novel KD framework that redefines distillation as a consensus space optimization task. This approach exploits the spontaneous convergence of high-level embeddings to scale-invariant, low-rank subspaces across model scales (see Fig. 1(d)). Specifically, we apply singular value decomposition on

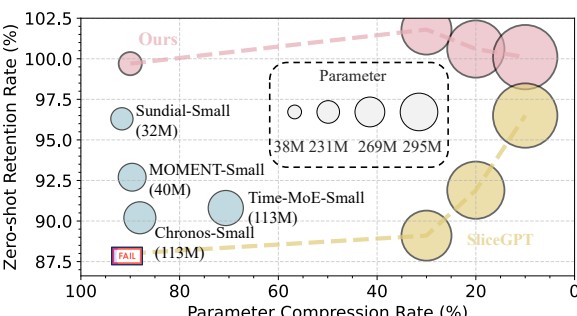

Figure 3: Benchmarked against MOMENT-Large, our method surpasses state-of-the-art approaches in zero-shot long-horizon forecasting retention and model compression. Unlike conventional methods that fail to significantly compress pre-trained models while preserving high-quality representations, our approach achieves a 90.13% parameter reduction. Even compared to the smallest mainstream time series foundation models, our compressed model delivers extreme compression while approximately preserving the original large-scale model performance.

consensus layers over local data subsets to extract low-rank subspaces and construct projection matrices. These guide student representations to align with the teacher's geometric centers, enabling efficient knowledge transfer without disrupting natural convergence structures. Unlike the rigid matching in feature distillation or the limitations in relation distillation, our method prioritizes subspace consensus over pointwise matching. It decouples from the teacher model, accelerating convergence and enhancing robustness across scales.

Our method also incorporates a scalable uncertainty injection mechanism to bridge biases from data selection. This models biases as frequency-domain gaps and enhances generalization to unseen data through inflated covariances. The design draws from two key insights. First, consensus spaces form spontaneously at varying depths, with deeper layers adding redundancy without altering geometric centers (Fig. 2(a-b)). Second, layer contributions are uneven, with shallow layers exhibiting zero-shot reconstruction capabilities and deeper layers showing long-tail redundancy (Fig. 1(c)). Our main contributions include:

- A new distillation perspective that transforms KD into a consensus space optimization problem, leveraging scale-invariant subspaces and hierarchical long-tail distributions for efficient TSFM compression.
- A method combining decomposition-based projection with frequency-domain uncertainty injection to align student representations and mitigate biases, ensuring stronger generalization.
- Extensive experiments on standard time series datasets, demonstrating the superiority of our model, where student models often outperform teacher models in prediction tasks.
- Our method achieves over 90% parameter reduction while retaining performance comparable to the base model, demonstrating advantages in compression ratio, performance retention, and distillation efficiency (Fig. 3).

## 2 RELATED WORK

### 2.1 EFFICIENT TIME SERIES FOUNDATION MODELING

Recent studies have emphasized Transformer-based time series foundation models (TSFMs) for handling complex data and enabling multi-task generalization (Liang et al., 2024). Masked reconstruction (MR), drawing inspiration from large language models, allows these models to learn robust representations. This approach works by randomly masking and reconstructing sequences, which helps capture contextual dependencies and long-term patterns (Liu et al., 2025). Notable examples include MOMENT, which uses multi-dataset pre-training to enhance data diversity (Goswami et al., 2024).

PatchTST employs patching and channel-independent processing to enable efficient long-sequence forecasting (Nie et al., 2023). Time-LLM reprograms large language models with MR and prompt engineering, thereby improving generalization (Jin et al., 2024). Additionally, Time-MoE scales up to 2.4 billion parameters, validating scaling laws across various domains (Shi et al., 2025). Despite these advances, the increasing model sizes create significant computational and storage challenges. This motivates our research on TSFM compression.

Various compression techniques exist, such as neural architecture search (NAS), pruning, knowledge distillation (KD), quantization, and low-rank mapping (Fournier et al., 2023). For instance, NAS methods like those proposed by Wang et al. optimize multivariate architectures but often require high computational resources and offer limited adaptability (Wang et al., 2024). Xu et al. introduced contrastive pruning to preserve task-agnostic knowledge through contrastive learning, which enhances generalization in pre-trained models (Xu et al., 2022). However, this approach overlooks high-level consensus in TSFMs. Quantization techniques, such as GPTQ (Frantar et al., 2023), perform well on proxy tasks but struggle with uncertainty calibration, which can impair long-context generalization (Li et al., 2024). Low-rank mapping reduces dimensions for key representations. Zha et al. (2024) applied decoupled spatio-temporal compression to high-dimensional data, yet it ignores hierarchical long-tail distributions, limiting its applicability to TSFMs. In comparison, KD effectively transfers knowledge from teacher to student models. This method better preserves TSFM consensus spaces and addresses hierarchical imbalances (Gao et al., 2024; Wu et al., 2021).

## 2.2 KNOWLEDGE DISTILLATION IN TSFMs

Knowledge distillation can be categorized into response-based methods, which imitate the teacher's soft label outputs (Hinton et al., 2015), feature-based methods, which align intermediate representations to capture deeper knowledge (Romero et al., 2014), and relation-based methods, which mine inter-instance or inter-layer relationships, such as similarity matrices, for robust transfer (Park et al., 2019a). For compressing Transformer-based TSFMs, several approaches have emerged. These include self-distillation frameworks that improve self-supervised efficiency through masked view prediction (Pieper et al., 2023), task-specific distillation with adversarial augmentation for downstream robustness (Zhang et al., 2022), and methods inspired by neural collapse or low-rank local feature distillation for better representation alignment (Zhang et al., 2025; Sy et al., 2025). Other techniques involve SliceGPT, which removes rows or columns from weight matrices to achieve up to 25% parameter reduction while preserving zero-shot performance (Ashkboos et al., 2024), FrameQuant's dynamic bit adjustment for balancing accuracy and efficiency (Adepu et al., 2024), probabilistic knowledge transfer for deep representation learning (Passalis & Tefas, 2018), relation mining (Park et al., 2019b), and contrastive distillation for enhanced alignment (Tian et al., 2020).

Existing methods often overlook the intrinsic low-rank consensus and uneven layer contributions in TSFM embeddings, leading to suboptimal compression (Ni et al., 2025; Zhao et al., 2024; Xu et al., 2023). We address this by reformulating distillation as consensus space optimization, leveraging inherent model structures to guide efficient compression.

## 3 METHODOLOGY

In this section, we introduce a novel consensus subspace distillation framework, as depicted in Fig. 1(d). This framework integrates scale-invariant low-rank representations, leveraging inherent model-agnostic structures observed across diverse time series foundation models (TSFMs), with hierarchical contribution screening to enable efficient compression. We then describe an extensible uncertainty injection mechanism to mitigate subset bias and enhance generalization. Finally, we outline the training procedure and associated losses.

### 3.1 CONSENSUS SUBSPACE DISTILLATION

**Offline Computation** For a given teacher model $\mathcal{T}$ and the full dataset $\mathcal{D} \in \mathbb{R}^{N \times C \times L}$, we first follow the central limit theorem to randomly sample an $n\%$ subset $\mathcal{D}_c \in \mathbb{R}^{\tilde{N} \times C \times L}$ for offline statistical estimation. In the offline phase, we first screen high-contribution layers: by setting all parameters except the $l$-th layer to zero, we define the teacher model variant $\theta_T^{[\forall i \neq l,\ i \rightarrow 0]}$ and compute the marginal contribution $\Delta \mathcal{L}^l = \mathcal{L}_{\text{MAE}}(\theta_T) - \mathcal{L}_{\text{MAE}}(\theta_T^{[\forall i \neq l,\ i \rightarrow 0]})$. We select the $K$ layer indices with the largest $\Delta \mathcal{L}^l$ as $\mathcal{I}_{\text{top}}$.

Next, we construct the consensus subspace: for each selected layer $l \in \mathcal{I}_{\text{top}}$, reshape the embedding $\bar{E}_T^l = \text{reshape}(E_T^l) \in \mathbb{R}^{H \times (\tilde{N}T)}$, where $E_T^l$ is the token embedding from the $l$-th layer of $\mathcal{T}$ (with $T$ tokens), and perform mean subtraction $\tilde{E}_T^l = \bar{E}_T^l - \frac{1}{\tilde{N}T}\bar{E}_T^l \mathbf{1}_{\tilde{N}T}\mathbf{1}_{\tilde{N}T}^\top$. Compute the average covariance

$$\Sigma_0 = \frac{1}{K}\sum_{l \in \mathcal{I}_{\text{top}}} \frac{1}{\tilde{N}T}\tilde{E}_T^l \tilde{E}_T^{l\top} \in \mathbb{R}^{H \times H}. \tag{1}$$

Since for any non-zero $v \in \mathbb{R}^H$, $v^\top \bar{\Sigma} v = v^\top \Sigma_0 v + \gamma \|v\|_2^2 \geq \gamma \|v\|_2^2 > 0$, to ensure positive definiteness, we apply shrinkage to obtain $\bar{\Sigma} = \Sigma_0 + \gamma I_H$ ($\gamma > 0$).

Then, perform $\bar{\Sigma} = U\Lambda U^\top$, where $\Lambda = \text{diag}(\lambda_1 \geq \cdots \geq \lambda_H)$. For rank truncation, select the rank based on the cumulative variance threshold $\theta$:

$$r = \min\left\{m : \frac{\sum_{i=1}^m \lambda_i}{\text{tr}\,\Lambda} \geq \theta\right\}, \quad U_r = [u_1, \ldots, u_r]. \tag{2}$$

The consensus projection matrix is $P_\mathcal{C} = U_r U_r^\top \in \mathbb{R}^{H \times H}$. For the token embedding $E_T^M$ from the top layer of $\mathcal{T}$, project $\bar{E}_T = P_\mathcal{C} E_T^M$, where $M$ is the total number of layers in $\mathcal{T}$. Reshape $Z = \text{reshape}(\bar{E}_T) \in \mathbb{R}^{H \times (\tilde{N}T)}$ and compute

$$\begin{cases} \mu_T = \dfrac{1}{\tilde{N}T}Z\mathbf{1}_{\tilde{N}T}, \\[2mm] \Sigma_T = \dfrac{1}{\tilde{N}T}(Z - \mu_T \mathbf{1}_{\tilde{N}T}^\top)(Z - \mu_T \mathbf{1}_{\tilde{N}T}^\top)^\top. \end{cases} \tag{3}$$

**Computation of Spectral Density** $S(\omega_k)$ In the offline computation phase, we obtain $S(\omega_k)$ for subsequent uncertainty injection. Based on the projected teacher embedding $\bar{E}_T \in \mathbb{R}^{\tilde{N} \times C \times T}$, first reshape it to $Z \in \mathbb{R}^{H \times (\tilde{N}T)}$. Perform FFT along the token dimension $T$ to obtain the frequency-domain embedding $E_{T,f} \in \mathbb{C}^{\tilde{N} \times C \times T}$. For each frequency point $\omega_k = \frac{2\pi k}{T}$ ($k = 0, 1, \ldots, T-1$), compute the frequency-domain covariance matrix of the embeddings:

$$\begin{aligned} S(\omega_k) = \frac{1}{\tilde{N}C}\sum_{n=1}^{\tilde{N}}\sum_{c=1}^{C}&\Big[\text{Re}(E_{T,f,n,c,k}) \cdot \text{Re}(E_{T,f,n,c,k})^\top \\ &+ \text{Im}(E_{T,f,n,c,k}) \cdot \text{Im}(E_{T,f,n,c,k})^\top\Big] \in \mathbb{R}^{H \times H}. \end{aligned} \tag{4}$$

The teacher cache provides the baseline spectral density $S(\omega_k) \in \mathbb{R}^{H \times H}$, which serves as the frequency-domain statistical benchmark of the teacher model on the subset $\mathcal{D}_c$, effectively capturing the covariance between spatial dimensions after embedding.

**Student Network Initialization** Copy the $K$ layers corresponding to $\mathcal{I}_{\text{top}}$ from the teacher $\mathcal{T}$ to the student $\mathcal{S}$, and add a low-rank increment only to the MLP output weights $W_{\text{mlp}}$ of the copied layers:

$$W_{\text{mlp}}^{\text{new}} = W_{\text{mlp}} + \mathbf{1}_{[r_a > 0]} \underbrace{AB}_{\text{rank } r_a}, \tag{5}$$

where $A \in \mathbb{R}^{H \times r_a}$, $B \in \mathbb{R}^{r_a \times H}$, and $r_a \ll H$. During training, freeze the original weights and only update $A$ and $B$ (if $r_a = 0$, it degenerates to an identity mapping).

**Remark:** The construction of the consensus subspace $\mathcal{C}$ ensures scale invariance, as dimensionality reduction captures the low-rank structure of embeddings, which spontaneously converges across different models, supporting efficient subspace alignment.

Next, we introduce the mean-covariance alignment loss for training consensus distillation.

**Mean-Covariance Alignment Loss** For the student top-layer embedding $E_S^K$, project $\bar{E}_S = P_\mathcal{C} E_S^K$, reshape $Z = \text{reshape}(\bar{E}_S) \in \mathbb{R}^{H \times (BT)}$, and compute

$$\begin{cases} \mu_S = \dfrac{1}{BT}Z\mathbf{1}_{BT}, \\[2mm] \Sigma_S = \dfrac{1}{BT}(Z - \mu_S \mathbf{1}_{BT}^\top)(Z - \mu_S \mathbf{1}_{BT}^\top)^\top. \end{cases} \tag{6}$$

For two Gaussian distributions $\mathcal{N}(\mu_T, \Sigma_T)$ and $\mathcal{N}(\mu_S, \Sigma_S)$, their squared 2-Wasserstein distance satisfies:

$$\mathcal{W}_2^2(\mathcal{N}_T, \mathcal{N}_S) = \underbrace{\|\mu_T - \mu_S\|_2^2}_{\mathcal{L}_\mu}$$
$$+ \underbrace{\mathrm{tr}\left(\Sigma_T + \Sigma_S - 2(\Sigma_T^{1/2}\Sigma_S\Sigma_T^{1/2})^{1/2}\right)}_{\Phi(\Sigma_S)}. \tag{7}$$

The covariance term $\Phi(\Sigma_S)$ has complex gradient computation, so we introduce a surrogate objective $g(\Sigma_S) = \|\Sigma_S - \Sigma_T\|_F^2$, as $g$ shares the global minimum with $\Phi$ at $\Sigma_S = \Sigma_T$. Thus, the objectives are $\mathcal{L}_\mu = \|\mu_S - \mu_T\|_2^2$ and $\mathcal{L}_\Sigma = g(\Sigma_S)$.

Our method unifies distillation, subspace projection, and mean-covariance alignment within a single framework, introducing a latent anchor manifold—rather than relying on a single teacher—based on new insights into scale-invariant convergence in TSFMs.

### 3.2 UNCERTAINTY INJECTION

We quantify the gap using frequency-domain ChF differences, converting it into spectral density inflation noise injected solely into the consensus subspace. This leverages uneven layer contributions to enhance conservatism without disrupting core representation centers, extending static alignment to dynamic uncertainty augmentation for improved generalization on unseen data.

Uncertainty computation directly uses the original input sequences $\mathcal{D} \in \mathbb{R}^{N \times C \times L}$ without requiring teacher forward passes, avoiding additional computational overhead. Perform fast Fourier transform (FFT) on the $L$ axis (time dimension) of the sequences to obtain the frequency-domain representation $x_f \in \mathbb{C}^{N \times C \times L}$ (complex tensor). Similarly, batch the subset $\mathcal{D}_c \in \mathbb{R}^{\tilde{N} \times C \times L}$ to obtain $\tilde{x}_f \in \mathbb{C}^{B \times C \times L}$. The frequency points $\omega_k$ are defined as the discrete frequencies in standard FFT: $\omega_k = \frac{2\pi k}{L}$, where $k = 0, 1, \ldots, L-1$.

**ChF Difference Quantification**    Inspired by Liang et al. (2024), we estimate the empirical characteristic function (ChF) to quantify the statistical differences between the two distributions $\mathcal{D}$ and $\mathcal{D}_c$ in the frequency domain. The ChF is the Fourier transform of the distribution, similar to a probability generating function, used to capture higher-order moment deviations. Computations are performed on the raw sequences, averaging across the spatial dimension $C$ to integrate multivariate information:

$$\Phi_x(\omega_k) = \frac{1}{NC} \sum_{n=1}^{N} \sum_{c=1}^{C} e^{i\,\mathrm{Re}(x_{f,n,c,k})}, \tag{8}$$

where $\Phi_x(\omega_k)$ represents the empirical ChF of samples $x$ in $\mathcal{D}$ at frequency $\omega_k$. Similarly, for samples $\tilde{x}$ in $\mathcal{D}_c$, $\Phi_{\tilde{x}}(\omega_k) = \frac{1}{BC} \sum_{b,c} e^{i\,\mathrm{Re}(\tilde{x}_{f,b,c,k})}$.

Then, compute the ChF difference to quantify the distance between the two ChFs:

$$\mathrm{Chf}(\omega_k) = |\Phi_x(\omega_k)|^2 + |\Phi_{\tilde{x}}(\omega_k)|^2$$
$$- 2|\Phi_x(\omega_k)||\Phi_{\tilde{x}}(\omega_k)| \cos(a_x - a_{\tilde{x}}), \tag{9}$$

where $|\Phi_x(\omega_k)|$ is the magnitude of $\Phi_x(\omega_k)$, and $a_x = \arg \Phi_x(\omega_k)$ is the argument. The global uncertainty $\mathcal{U}$ aggregates differences across all frequencies:

$$\mathcal{U} = \sum_{k=0}^{L-1} \mathrm{Chf}(\omega_k) w(\omega_k), \quad w(\omega_k) = \exp\left(-\omega_k^2/2\sigma^2\right), \tag{10}$$

where $w(\omega_k)$ is a Gaussian weighting function.

**Spectral Density Inflation**    First, based on the ChF differences, construct a scaling factor (inflation factor):

$$\gamma(\omega_k) = 1 + \lambda \frac{\mathrm{Chf}(\omega_k)}{|\Phi_x(\omega_k)|^2 + \varepsilon}, \quad 0 < \lambda \le 1. \tag{11}$$

Then, compute the inflated spectral density $S_\star(\omega_k) = \gamma(\omega_k)S(\omega_k)$. Further, obtain the zero-lag covariance:

$$\Sigma_\star = \frac{1}{T}\sum_{k=0}^{T-1} S_\star(\omega_k). \tag{12}$$

If $\mathcal{U} \to 0$, then $\Sigma_\star \to \Sigma_T$.

**Consensus Space Injection** Project $\Sigma_\star^{\mathcal{C}} = U_r^\top \Sigma_\star U_r \in \mathbb{R}^{r \times r}$. Perform eigenvalue decomposition on $\Sigma_\star^{\mathcal{C}}$ to get $\Sigma_\star^{\mathcal{C}} = V_\star \Lambda_\star V_\star^\top$, where $\Lambda_\star = \mathrm{diag}(\lambda_{\star,1} \geq \cdots \geq \lambda_{\star,r})$ is the diagonal matrix of eigenvalues. Then, compute the gain coefficients:

$$\Gamma = \mathrm{diag}\left(\sqrt{\frac{\lambda_{\star,1}}{\lambda_1}}, \ldots, \sqrt{\frac{\lambda_{\star,r}}{\lambda_r}}\right), \tag{13}$$

where $\lambda_i$ are from the original consensus subspace's $\Lambda$. The final uncertainty-injected consensus projection matrix is $P_{\mathcal{C}}^\gamma = U_r \Gamma U_r^\top$. These formulations, though complex, precisely capture scale-invariant subspaces, supported by empirical convergence in Fig. 2.

### 3.3 Overall Procedure

In the offline phase, precompute $P_{\mathcal{C}}$, $(\mu_T, \Sigma_T)$, $\{S(\omega_k)\}$, $\Sigma_\star$, and $\Gamma$. In the online phase, the student computes $E_S^K$, obtains $\bar{E}_S = P_{\mathcal{C}}^\gamma E_S^K$ via the uncertainty-injected consensus projection, and feeds it into the mean-covariance alignment loss. The overall loss is $\mathcal{L} = \mathcal{L}_{\text{task}} + \beta(\mathcal{L}_\mu + \mathcal{L}_\Sigma)$, with gradients backpropagated naturally. The algorithmic procedure is summarized in Algorithm 1.

---

**Algorithm 1** Consensus Subspace Distillation Framework

1: **Input:** $\mathcal{T}, \mathcal{D}, n\%, K, \theta, \lambda, \sigma, \beta$.
2: **Output:** $\mathcal{S}$.
3: **Offline:**
4: Sample $\mathcal{D}_c$ from $\mathcal{D}$.
5: Select $\mathcal{I}_{\text{top}}$ by $\Delta\mathcal{L}^l$.
6: Build $P_{\mathcal{C}}$ from $\bar{\Sigma}$.
7: Compute $(\mu_T, \Sigma_T)$, $S(\omega_k)$, $\mathcal{U}$.
8: Inflate to $P_{\mathcal{C}}^\gamma$.
9: **Init:** Copy top layers to $\mathcal{S}$.
10: **Online:**
11: **while** not converged **do**
12:    Get $E_S^K$, project to $\bar{E}_S$.
13:    Compute losses: $\mathcal{L} = \mathcal{L}_{\text{task}} + \beta(\mathcal{L}_\mu + \mathcal{L}_\Sigma)$.
14:    Update $\mathcal{S}$.
15: **end while**

---

## 4 Experiment Results

### 4.1 Dataset and Model Setup

**Dataset Selection** We evaluate the proposed consensus subspace distillation framework on the Time Series Pile, a diverse collection of approximately 13 million time series spanning 13 distinct real-world domains, including healthcare, electricity, economics, and transportation, with a total of 1.23 billion timestamps (Goswami et al., 2024). This dataset ensures comprehensive validation of the model's cross-domain generalization capabilities. Following the experimental setup of MOMENT, we select datasets for long-term forecasting, imputation, classification, and anomaly detection tasks. We adopt standard dataset splits and preprocessing procedures, with detailed metadata and splitting methods available in our code repository.

**Baselines and Teacher Model.** We use MOMENT-Large as the teacher model (Goswami et al., 2024), pretrained on the Time Series Pile through masked reconstruction. The baselines include state-of-the-art knowledge distillation methods: Probabilistic Knowledge Transfer (PKT) (Passalis & Tefas, 2018), Relational Knowledge Distillation (RKD) (Park et al., 2019b), Contrastive Representation Distillation (CRD) (Tian et al., 2020), Adversarial Data Augmentation for KD (ADA-KD) (Zhang et al., 2022), Low-Rank Local Feature Distillation (LRLFD) (Sy et al., 2025), and Neural Collapse Inspired KD (NCKD) (Zhang et al., 2025). We also include SliceGPT (Ashkboos et al., 2024), which supports low-rank mapping and pruning, along with the quantization method FrameQuant (Adepu et al., 2024).

### 4.2 Implementation Details

All experiments use 8 NVIDIA RTX 4090 GPUs, with distillation on one 4090 GPU for 3 epochs. We copy the top $K = 3$ layers from the teacher model, selected via marginal contribution screening, and

add a low-rank increment to MLP output weights with rank $r_a = 64$. In the offline phase, a $n = 10\%$ dataset subset is sampled to compute the consensus subspace projection matrix $P_C$, using a truncation threshold of $\theta = 0.99$ and spectral density inflation ($\lambda = 0.1$, $\sigma = 1.0$). In the online phase, we use the AdamW optimizer (learning rate $1 \times 10^{-4}$, batch size $B = 2048$) and loss weighting $\beta = 0.5$. Full implementation details, including random seeds, training configurations, ablation study protocols, and model weights, are available in our public code repository.

## 4.3 COMPARE WITH SOTA METHODS

Table 1: Comparison of different model compression methods. Results are averaged over multiple datasets for each task. Long-Horizon Forecasting uses average MAE over 8 datasets with forecast horizons $\{96, 192, 336, 720\}$; Imputation uses average MAE over 6 datasets with mask ratios $\{0.125, 0.25, 0.375, 0.5\}$; Anomaly Detection uses average Adj. Best F1 and VUS-ROC over 248 datasets; Classification uses average accuracy over 91 datasets.

| Method | Long-Horizon Forecasting $_{LP}$ | Imputation $_{LP}$ | Anomaly Detection $_{LP}$ | | Classification $_0\uparrow$ | Comp. Time $\downarrow$ | Model Parameters | Ref. |
| | MAE (Avg.) $\downarrow$ | MAE (Avg.) $\downarrow$ | Adj. Best F1 $\uparrow$ | VUS-ROC $\uparrow$ | | | | |
| Teacher | 0.476 | 0.159 | 0.721 | 0.728 | 0.764 | N/A | 385M | ICML'24 |
| **Ours** | **0.471 (+1.05%)** | **0.157 (+1.26%)** | **0.713 (-1.11%)** | **0.721 (-0.96%)** | **0.633 (-17.01%)** | ∼28 h | **38M (-90.13%)** | - |
| SliceGPT | 0.528 (-10.92%) | 0.176 (-10.69%) | 0.630 (-12.62%) | 0.647 (-11.13%) | 0.685 (-10.34%) | ∼10 s | 274M (-28.83%) | ICLR'24 |
| FrameQuant | 0.585 (-22.90%) | 0.218 (-37.11%) | 0.541 (-25.00%) | 0.572 (-21.43%) | 0.580 (-24.08%) | ∼3 h | 385M (15x Mem. ↓) | ICML'24 |

(Reduced)

Table 2: Comparison of different distillation methods under 9.87% parameter retention (student model). We can achieve promising results using only 10% of the data for distribution distillation. Results are averaged over multiple datasets for each task. Long-Horizon Forecasting uses MAE averaged over 8 datasets with forecast horizons 96, 720. Imputation uses MAE averaged over 6 datasets with mask ratios 0.125, 0.500. Anomaly Detection uses Adj. Best F1 and VUS-ROC averaged over 248 datasets. Classification uses accuracy averaged over 91 datasets. Distillation Time indicates the time taken for the distillation process.

| Method | Long-Horizon Forecasting $_{LP}$ | | Imputation $_{LP}$ | | Anomaly Detection $_0$ | | Classification $_0\uparrow$ | Distillation Time (GPU h) $\downarrow$ | Ref. |
| | MAE (96) $\downarrow$ | MAE (720) $\downarrow$ | MAE (0.125) $\downarrow$ | MAE (0.500) $\downarrow$ | Adj. Best F1 $\uparrow$ | VUS-ROC $\uparrow$ | | | |
| Teacher | 0.299 | 0.381 | 0.159 | 0.158 | 0.569 | 0.660 | 0.764 | N/A | ICML'24 |
| PKT | 0.418 (-39.80%) | 0.514 (-34.91%) | 0.203 (-27.67%) | 0.229 (-44.94%) | 0.478 (-16.00%) | 0.561 (-15.00%) | 0.550 (-28.01%) | 204.73 | ECCV'18 |
| RKD | 0.368 (-23.08%) | 0.457 (-19.95%) | 0.183 (-15.09%) | 0.198 (-25.32%) | 0.512 (-10.02%) | 0.594 (-10.00%) | 0.588 (-23.04%) | 180.19 | CVPR'19 |
| CRD | 0.344 (-15.05%) | 0.434 (-13.91%) | 0.171 (-7.55%) | 0.187 (-18.35%) | 0.535 (-5.98%) | 0.620 (-6.06%) | 0.610 (-20.16%) | 220.46 | ICLR'20 |
| ADA-KD | 0.359 (-20.07%) | 0.465 (-22.05%) | 0.190 (-19.50%) | 0.205 (-29.75%) | 0.524 (-7.91%) | 0.607 (-8.03%) | 0.595 (-22.12%) | 190.82 | AAAI'22 |
| LRLFD | 0.353 (-18.06%) | 0.449 (-17.85%) | 0.175 (-10.06%) | 0.193 (-22.15%) | 0.518 (-8.96%) | 0.600 (-9.09%) | 0.602 (-21.20%) | 210.37 | NAACL'25 |
| NCKD | 0.347 (-16.05%) | 0.442 (-16.01%) | 0.179 (-12.58%) | 0.200 (-26.58%) | 0.529 (-7.03%) | 0.613 (-7.12%) | 0.605 (-20.81%) | 195.64 | AAAI'25 |
| **Ours** | **0.287 (+4.01%)** | **0.366 (+3.94%)** | **0.152 (+4.40%)** | **0.151 (+4.43%)** | **0.557 (-2.11%)** | **0.647 (-1.97%)** | **0.633 (-17.15%)** | **3.86** | - |

(Student)

Table 3: Zero-shot long-horizon forecasting comparison between our compressed model and mainstream time series foundation models. Corresponding prediction lengths include $\{96, 192, 336, 720\}$. Averaged results of four prediction lengths are reported here. 1[st] Count refers to the number of datasets where the current model attains the top-ranked average performance over all forecasting horizons. Results of baseline models are officially reported by Liu et al. (2025). Datasets in pre-training are not evaluated on corresponding models, which are denoted by the dash ($-$).

| Models | Ours | | MOMENT$_{Small}$ | | MOMENT$_{Large}$ | | Time-MoE$_{Base}$ | | Time-MoE$_{Large}$ | | Time-MoE$_{Ultra}$ | | Sundial$_{Small}$ | | Sundial$_{Base}$ | | Sundial$_{Large}$ | | Chronos$_{Base}$ | | Chronos$_{Large}$ | |
| | - | | ICML'24 | | ICML'24 | | ICLR'25 | | ICLR'25 | | ICLR'25 | | ICML'25 | | ICML'25 | | ICML'25 | | TMLR'24 | | TMLR'24 | |
| Metric $\downarrow$ | MSE | MAE | MSE | MAE | MSE | MAE | MSE | MAE | MSE | MAE | MSE | MAE | MSE | MAE | MSE | MAE | MSE | MAE | MSE | MAE | MSE | MAE |
| ETTm1 | 0.356 | 0.381 | 0.354 | 0.391 | 0.345 | 0.380 | 0.394 | 0.415 | 0.376 | 0.405 | 0.356 | 0.391 | 0.354 | 0.388 | 0.336 | 0.377 | 0.331 | 0.369 | 0.645 | 0.500 | 0.555 | 0.465 |
| ETTm2 | 0.262 | 0.318 | 0.269 | 0.325 | 0.260 | 0.319 | 0.317 | 0.365 | 0.316 | 0.361 | 0.288 | 0.344 | 0.265 | 0.324 | 0.258 | 0.320 | 0.254 | 0.315 | 0.310 | 0.350 | 0.295 | 0.338 |
| ETTh1 | 0.403 | 0.436 | 0.427 | 0.442 | 0.419 | 0.435 | 0.400 | 0.424 | 0.394 | 0.419 | 0.412 | 0.426 | 0.390 | 0.418 | 0.411 | 0.434 | 0.395 | 0.420 | 0.591 | 0.468 | 0.588 | 0.466 |
| ETTh2 | 0.351 | 0.397 | 0.358 | 0.411 | 0.353 | 0.395 | 0.366 | 0.404 | 0.405 | 0.415 | 0.371 | 0.399 | 0.340 | 0.387 | 0.333 | 0.387 | 0.334 | 0.387 | 0.405 | 0.410 | 0.455 | 0.427 |
| ECL | 0.169 | 0.270 | 0.171 | 0.264 | 0.166 | 0.261 | - | - | - | - | - | - | 0.169 | 0.265 | 0.169 | 0.265 | 0.166 | 0.262 | 0.214 | 0.278 | 0.204 | 0.273 |
| Weather | 0.226 | 0.269 | 0.237 | 0.277 | 0.227 | 0.268 | 0.265 | 0.297 | 0.270 | 0.300 | 0.256 | 0.288 | 0.233 | 0.271 | 0.234 | 0.270 | 0.238 | 0.275 | 0.292 | 0.315 | 0.279 | 0.306 |
| 1[st] Count | 37 | 36 | 26 | 23 | 40 | 44 | 16 | 16 | 16 | 16 | 19 | 22 | 41 | 40 | 43 | 40 | 48 | 48 | 3 | 5 | 7 | 8 |

**Compression Performance Comparison** We benchmark our consensus subspace optimization method against MOMENT-Large (385M parameters) and other state-of-the-art techniques, such as SliceGPT and FrameQuant (Tab. 1). Our approach delivers comparable performance in forecasting and imputation, while retaining over 90% accuracy in anomaly detection and classification tasks. Our method compresses the model to 38M parameters, achieving a 90.13% reduction that significantly surpasses the 28.83% from SliceGPT. By extracting scale-invariant low-rank subspaces, our technique overcomes rigid matching challenges and validates that consensus spaces serve as effective manifolds for efficient knowledge transfer.

**Distillation Method Comparison** Tab. 2 shows our method outperforms prior distillation techniques using only 9.87% of the parameters and 3.86 GPU hours of training. It improves upon the teacher model with a 4.01% MAE reduction in forecasting and a 4.40% reduction in imputation, while maintaining robust performance in anomaly detection and classification. These results demonstrate that our low-rank alignment to geometric centers enables efficient and bias-resistant knowledge transfer.

| Method | Anomaly Detection $_0$ | | Classification $_0\uparrow$ |
|---|---|---|---|
| | Adj. Best F1 $\uparrow$ | VUS-ROC $\uparrow$ | |
| Teacher | 0.723 | 0.726 | 0.767 |
| Ours ($K$=4) | 0.651 (-9.89%) | 0.662 (-9.38%) | 0.704 (-8.36%) |
| Ours ($K$=4, w/o UI) | 0.645 (-10.79%) | 0.656 (-10.19%) | 0.698 (-9.00%) |
| Ours ($K$=8) | 0.717 (-1.54%) | 0.721 (-1.16%) | 0.752 (-1.87%) |
| Ours ($K$=8, w/o UI) | 0.712 (-1.94%) | 0.716 (-1.93%) | 0.747 (-2.48%) |
| Ours ($K$=12) | 0.728 (-0.12%) | 0.729 (-0.43%) | 0.761 (-0.54%) |
| Ours ($K$=12, w/o UI) | 0.724 (-0.69%) | 0.725 (-0.96%) | 0.757 (-1.17%) |
| Ours ($K$=16) | 0.716 (-0.89%) | 0.727 (-0.81%) | 0.758 (-1.12%) |
| Ours ($K$=16, w/o UI) | 0.711 (-1.66%) | 0.722 (-1.52%) | 0.753 (-1.83%) |

Table 4: Ablation study on the number of layers (K) in our method, including cases without uncertainty injection (UI).

**Foundation Model Comparison** The analysis in Tab. 3 demonstrates that our compressed model sustains competitive performance against larger time series foundation models. It excels in datasets like ETTm1 and ETTh2 with lower average MSE and MAE. These results emphasize balanced retention of temporal dynamics through subspace alignment, even after significant parameter reduction.

### 4.4 ABLATION STUDY

Tab. 4 presents our ablation on the number of layers ($K$). Higher values, such as $K = 12$, produce student models that nearly match the teacher's performance in anomaly detection and classification. This indicates that deeper hierarchical integration better captures essential consensus subspaces. Variants with uncertainty injection (UI) consistently outperform those without, particularly at lower $K$, by compensating for representational biases and enabling robust knowledge transfer. The trend suggests an optimal $K$ around 12, where further increases add redundancy without proportional benefits in alignment to scale-invariant geometric centers.

### 4.5 DOES CONSENSUS-SPACE DISTILLATION PRESERVE DISCRIMINATIVE REPRESENTATIONS?

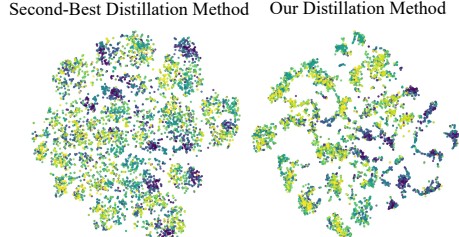

Figure 4: PCA and t-SNE visualizations of the representations learned by our method and other baselines on the Crop datasets, with distinct colors indicating different classes. Without dataset-specific fine-tuning, our method produces separable, clustered representations, indicating potential for effective feature extraction in downstream classification.

Fig. 4 visualizes the learned representations on the large-scale Crop classification dataset, comparing our method with CRD, the second-best approach. Our method yields more distinct and well-separated class representations, even in a zero-shot setting without ground-truth labels. The representation is from the distilled model's final layer output.

## 5 CONCLUSION

We reformulate knowledge distillation for Transformer-based time series foundation models as a consensus subspace optimization problem, exploiting the convergence of high-level embeddings to scale-invariant, low-rank subspaces. Our framework achieves 90.13% parameter reduction and 100x distillation speedup while maintaining performance across zero-shot tasks including forecasting, imputation, anomaly detection, and classification. Experiments demonstrate superiority over state-of-the-art methods, enabling efficient deployment in resource-limited settings.

### LLM USAGE STATEMENT

Large language models (LLMs) were used only for grammar and style editing. All research and content were created by the authors. The authors take full responsibility for the paper's content.

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
