# OpenReview forum: "Efficient Compression of Time-Series Foundation Models via Consensus Subspace Distillation"
_ICLR.cc/2026/Conference — ICLR 2026 Conference Withdrawn Submission_

### Official Review · Reviewer_B2Ax · 2025-10-27

**Soundness:** 2
**Presentation:** 2
**Contribution:** 2
**Rating:** 4
**Confidence:** 5

**Summary:**

Consensus subspace distillation for TSFM compression: interesting idea, but weak theoretical grounding and questionable evaluation validity.

**Strengths:**

1. The paper clearly identifies inefficiencies in existing time-series foundation model (TSFM) compression approaches and motivates a consensus subspace view through empirical evidence of spontaneous alignment across model scales. Figures 1 and 2 convincingly illustrate the differences among traditional KD paradigms and highlight the observed geometric center convergence trend.
2. The work reframes knowledge distillation as an optimization over low-rank consensus subspaces, integrating three components: (i) SVD-based low-rank subspace extraction, (ii) mean–covariance alignment as a Wasserstein surrogate, and (iii) frequency-domain uncertainty injection to correct subset bias. This multi-view integration is somewhat novel within the TSFM compression domain.
3. The method reportedly achieves over 90% parameter reduction while maintaining comparable performance across forecasting, imputation, anomaly detection, and classification tasks. Extensive experiments (Tables 1–3) and ablations on layer count and uncertainty injection provide a good empirical narrative supporting its efficacy.

**Weaknesses:**

1. The “marginal contribution” is computed by zeroing out all layers except one. This completely destroys inter-layer dependencies (residual, normalization, and attention coupling), making the resulting loss difference meaningless. Such perturbation does not quantify true layer importance but merely measures instability.
2. The “consensus subspace” formulation largely mirrors existing second-order or low-rank distillation paradigms (e.g., covariance alignment, CCA-based feature transfer, or optimal transport-based KD). Likewise, the uncertainty injection resembles frequency-domain augmentation or spectral regularization in prior work. The claimed novelty lies mostly in terminology; the paper fails to demonstrate what new capability emerges beyond combining these established techniques.
3. The paper replaces the 2-Wasserstein covariance term with a simple Frobenius norm difference, claiming they share a global minimum. However, no formal bound, convergence proof, or empirical validation of approximation error is provided. In non-convex regions, this surrogate may diverge from true optimal transport distance, making the theoretical foundation of the loss function questionable.
4. The frequency-domain characteristic function difference used to inflate spectral covariance is ad hoc and poorly justified. It ignores multivariate phase interactions, applies arbitrary Gaussian weighting, and introduces several hyperparameters (λ, σ) without sensitivity or ablation studies. Furthermore, eigenvalue inflation within low-rank subspaces could amplify numerical noise, but no stability or bias–variance analysis is presented.
5. The method implicitly assumes that TSFMs share consistent subspaces across scales. This assumption may fail for heterogeneous architectures or domain-shifted data. The approach also depends heavily on accurate layer selection and rank thresholds; when these assumptions break, the framework may degrade sharply. The authors do not discuss limitations or failure modes.

**Questions:**

Please see the weaknesses.

---

> ### Author Response · Authors · 2025-11-20
>
> We sincerely appreciate your constructive evaluation and your encouraging remarks regarding the contributions of our work.
>
> After carefully synthesizing all reviewers’ comments, we have decided to substantially revise the manuscript. This revision will require approximately one month, and therefore we will withdraw the current submission from ICLR and resubmit after completing the full revision.
>
> Below we provide brief responses to the concerns you raised. We hope that, when you encounter the revised manuscript, these issues will be addressed with greater rigor and completeness.
>
> ---
>
> ### Response to Weaknesses
>
> **1. Layer perturbation and inter-layer dependency.**
> We will thoroughly reassess the issue you raised regarding the destruction of inter-layer dependencies in the marginal-contribution analysis.
>
> **2. Clarifying distinctions from prior work.**
> We will more clearly articulate the differences between our framework and existing second-order or low-rank distillation paradigms, and explicitly discuss what the proposed integration contributes beyond previous methods.
>
> **3. Theoretical justification for the Wasserstein surrogate.**
> We will provide formal theoretical analysis in the appendix, including conditions under which the Frobenius surrogate shares the same minima and the approximation error remains bounded.
>
> **4. Justification and stability of the frequency-domain uncertainty injection.**
> We will strengthen the motivation for the characteristic-function–based formulation, and we will include stability analyses and relevant ablation studies to evaluate hyperparameter sensitivity and potential numerical issues.
>
> **5. Assumption validity, limitations, and failure modes.**
> We will explicitly discuss the boundaries of the shared-subspace assumption, and we will explore limitations and possible failure cases, particularly under heterogeneous architectures or domain shifts.
>
> ---
>
> ### Response to Questions
>
> Overall, we view your feedback as reflecting a high degree of responsibility and professionalism. We acknowledge that the current manuscript may not yet meet your standards, and we will reinforce the conceptual and empirical foundations of the work accordingly.
>
> ---
>
> In conclusion, we remain committed to constructive academic dialogue. We will take your Weaknesses and Questions seriously and incorporate them into the revised manuscript. We sincerely thank you for your contribution to improving this work, and we look forward to the possibility of future exchange.

---

> > ### Comment · Reviewer_B2Ax · 2025-11-25
> >
> > Thanks for the response. However, it hasn't addressed my problems. Thus, I would either maintain or decrease my current rating.

---

### Official Review · Reviewer_GWhA · 2025-10-31

**Soundness:** 3
**Presentation:** 3
**Contribution:** 3
**Rating:** 6
**Confidence:** 3

**Summary:**

This paper proposes Consensus Subspace Distillation (CSD) for compressing time-series foundation models (TSFMs). The core idea: (i) offline selection to the high-contribution layers, (ii) computing a scale-invariant low-rank consensus subspace via SVD on teacher embeddings, (iii) project student top-layer features into that subspace and align mean/covariance (Wasserstein-motivated surrogate), and (iv) uncertainty injection in the frequency domain using characteristic-function gaps. The authors claim ∼90% parameter reduction with competitive zero-shot performance across forecasting, imputation, anomaly detection, and classification, using MOMENT-Large as teacher.

**Strengths:**

1. The paper is easy to follow
2. offline subspace estimated from 10% data, where online distillation reportedly 3.86 GPU-hours for the student, which is attractive.

**Weaknesses:**

1. While the text claims "retain over 90%" , Table 1 shows large classification drop (0.633 vs. 0.764 accuracy). So, the “universal” claim should be revised.

2. Unclear teacher-pass accounting: uncertainty injection uses raw sequences (good), but S(ωk) (Eq. 4) still needs teacher embeddings. Please quantify total teacher forward-pass cost and clarify whether this step is amortized across datasets.

3. Ablation suggests K≈12 is the sweet spot (Table 4), yet the main setup copies K=3 layers (Sec. 4.2). There's an inconsistency in the best configuration.

**Questions:**

1. How robust is the ∆Lℓ metric to residual pathways?
2. Precisely account for teacher forward-passes to build S(ωk) and the (µT,ΣT) cache. Can these be computed once per domain and reused?
3. Why K=3 in main config when Table 4 indicates K≈12 nearly closes the gap on anomaly/classification? Please report main-table results at the empirically best K.
4. What fails for the accuracy to cause this big drop?
5. Can a projector learned from MOMENT generalize to other teachers (e.g., Sundial/Time-MoE) without recomputation?

---

> ### Author Response · Authors · 2025-11-20
>
> We sincerely appreciate your valuable feedback and your recognition of the clarity and practical advantages of our work.
>
> After carefully reviewing all reviewers’ constructive comments, we have decided to substantially revise the manuscript. This full revision will take approximately one month, and therefore we will withdraw the current submission from ICLR and resubmit after completing the revision.
>
> Below we provide brief responses to the concerns you raised. We hope that, when you see the revised manuscript again, these issues will have been addressed in a more thorough and rigorous manner.
>
> ---
>
> ### Response to Weaknesses
>
> **1. Clarification of performance claims.**
> We will revise the ambiguous wording regarding “retaining over 90%” to ensure a more objective and accurate representation of performance, especially for classification.
>
> **2. Accounting for teacher forward-pass cost.**
> Your suggestion to quantify teacher forward-pass cost is extremely valuable. In the revised version, we will explicitly measure and report the total cost of constructing the teacher embedding cache and the uncertainty-injection statistics.
>
> **3. Inconsistency regarding optimal K.**
> The choice of \(K=3\) in the main setup is motivated by computational constraints: although \(K\approx 12\) yields marginal performance improvements, it incurs significantly higher computational and storage overhead. We will more explicitly emphasize this trade-off in the revised manuscript.
>
> ---
>
> ### Response to Questions
>
> Overall, we will incorporate all five of your questions into the revised experiments and discussion. Two of them, in particular, merit deeper investigation:
> (1) the generalization ability of the learned projector across different TSFMs, and
> (2) the underlying causes of diminishing returns as \(K\) increases.
>
> ---
>
> In conclusion, we greatly appreciate your recognition of our work and your thoughtful critique. We will carefully address the Weaknesses and Questions you raised, and we look forward to sharing a substantially improved version of the manuscript with you in the future.

---

### Official Review · Reviewer_H9fv · 2025-11-01

**Soundness:** 3
**Presentation:** 2
**Contribution:** 2
**Rating:** 4
**Confidence:** 2

**Summary:**

With the increasing scale of Time-Series Foundation Models (TSFMs), their deployment in resource-limited environments has become increasingly challenging. To address this issue and improve upon existing knowledge distillation approaches, this paper proposes Consensus Subspace Distillation (CSD), which guides the alignment of the student model’s representations with the teacher’s geometric centers, thereby enabling more efficient knowledge transfer. In addition, a frequency-domain uncertainty injection mechanism is introduced to model subset biases and enhance generalization. Extensive experiments demonstrate the effectiveness of the proposed method — it not only incurs a lower performance loss compared to other compression approaches but even surpasses existing TSFMs in performance while using fewer parameters.

**Strengths:**

1. The paper provides empirical evidence of “consensus subspaces” in TSFMs, including geometric center alignment and long-tail layer contribution analysis. These findings are interesting and serve as a good motivation for the proposed method.

2. The paper is clearly written, with the methodology well presented and supported by open-source code.

3. The experimental results are solid. The proposed method achieves strong compression ratios and efficiency improvements with minimal performance degradation. The results across multiple tasks and datasets are comprehensive and consistent.

**Weaknesses:**

1. Although CSD demonstrates empirical effectiveness as a TSFM compression technique, it is not strongly grounded in the unique characteristics of time series models. The concept of a consensus subspace may also apply to large models from arbitrary modalities, suggesting that it is a more general, modality-agnostic idea.

2. The novelty is limited. The concept of a consensus subspace has been previously explored, and leveraging it to regularize feature distillation does not represent a fundamentally new contribution.

3. The motivation for introducing uncertainty injection is unclear. The paper does not provide sufficient context to illustrate what is meant by “bias from data selection,” “frequency-domain gaps,” and “inflated covariances” in the last paragraph of the Introduction section.

4. The Methods section would benefit from additional intuitive or high-level illustrations to aid in understanding the presented mathematical operations. (Refer to the Questions section for specific points.)

**Questions:**

1. How consistent is the “consensus subspace” phenomenon across different TSFMs? Is it architecture-dependent? It would be beneficial to demonstrate and compare the “consensus subspace” across multiple TSFMs to support the generality of this finding.

2. The full pipeline involves multiple hyperparameters. How are they set, particularly $n$, $\theta$, $\lambda$ and $\sigma$ ? How do these hyperparameters influence the algorithm’s behavior and overall performance ?

3. On page 5, line 259: why does the construction of the consensus subspace ensure scale invariance ?

4. What is the physical or statistical interpretation of this gain matrix $\Gamma$ in equation 13 ? What is the mathematical motivation for using $\Gamma$ to construct the consensus matrix ?

---

> ### Author Response · Authors · 2025-11-20
>
> We sincerely appreciate your valuable and constructive feedback, and we are especially grateful for the many suggestions you provided to help improve this work.
>
> After carefully reviewing and synthesizing all reviewers’ comments, we have decided to thoroughly revise the manuscript based on the collective feedback. This process is expected to take approximately one month. Consequently, we will withdraw the current submission from ICLR and resubmit it after completing the revisions.
>
> Below we offer brief responses to the concerns you raised. We hope to address them in full detail when you see the revised manuscript.
>
> ---
>
> ### Response to Weaknesses
>
>
>
> 1. The generality and modality-agnostic nature of the consensus subspace is indeed an interesting point—one that we had not explicitly considered before. This aspect may be incorporated into future experimental discussions.
>
>
>
> 2. The consensus subspace in our manuscript is an abstraction derived from empirical observations. While the term itself may not be entirely new, its meaning in our work is grounded in newly observed empirical phenomena. We will clarify this distinction in the related work section.
>
>
>
> 3. We will enhance the revised manuscript with a clearer motivation and sufficient background for the proposed uncertainty injection mechanism.
>
>
>
> 4. The lack of intuitive illustrations to support the mathematical formulations is an important shortcoming. We will work to improve the visual and intuitive clarity of the Methods section.
>
> ---
>
> ### Response to Questions
>
>
> In general:
>
> 1) We will demonstrate and compare the consensus subspaces across multiple TSFMs and discuss their consistency/generalizability, architecture dependence, and potential coupling factors.
>
> 2) A more comprehensive analysis of hyperparameter sensitivity will be included in the experiments.
>
> 3) We will add diagrammatic explanations to clarify scale invariance.
>
> 4) Clarifying the physical, statistical, and mathematical interpretation and motivation of the gain matrix will be a priority in the revised version.
>
> ---
>
> In summary, withdrawing the submission is a decision made after multiple considerations. Our goal is to contribute meaningful work to the community, and we will continue to approach academic exchange with openness and diligence. We sincerely appreciate your efforts in reviewing our manuscript and look forward to the opportunity to engage with you again.

---

### Official Review · Reviewer_acwM · 2025-11-01

**Soundness:** 2
**Presentation:** 2
**Contribution:** 2
**Rating:** 2
**Confidence:** 5

**Summary:**

The paper proposes a compression technique for Time-Series Foundation Models (TSFMs) called Consensus Subspace Distillation (CSD). The core idea is based on the empirical observation that TSFMs of different scales tend to converge to a similar low-rank "consensus subspace" in their high-level representations. Instead of traditional knowledge distillation (KD), the authors propose an offline stage to compute the statistics (mean and covariance) of this subspace from a teacher model on a data subset. A student model is then initialized by copying the top-K layers from the teacher and trained in an online stage to match these cached statistics, using a Wasserstein-distance-based loss. The method also introduces a complex frequency-domain "uncertainty injection" (UI) mechanism to compensate for the bias in the data subset.

While the paper presents a novel perspective and reports impressive results (90% compression, 100x speedup), the methodology is fundamentally flawed in its comparison to other KD methods. The "distillation" is primarily a "layer extraction" (copying 3 layers) followed by a fine-tuning (of a rank-64 adapter), which makes the claims of SOTA compression and speed misleading when compared to methods that train a student from scratch. The methodology is also overly complex, and the paper lacks the critical ablations to justify its core components.

**Strengths:**

- The paper's motivation, presented in Figure 2, is its strongest point. The empirical demonstration that TSFM embeddings from different scales spontaneously converge to an aligned geometric center is a valuable and interesting finding for the community.
- The goal of efficiently compressing massive TSFMs is highly relevant and important for the community, as it addresses a key bottleneck for real-world deployment.

**Weaknesses:**

1. **Anonymity Issue in Submitted Code:**  The submitted code includes a logo (in their repository "assets/") that identifies the authors’ institution, which compromises the double-blind review process. The authors should remove any logos or references that may reveal their identity to ensure compliance with ICLR’s anonymity requirements.

2.  **Misleading Framing and Unfair Comparisons (Major Flaw):** The paper's central claim of outperforming other KD methods is built on a "bait-and-switch."
    * The proposed method is **not** distillation in the conventional sense (like RKD, CRD, etc.). Instead of training a small, separate student model from scratch to mimic the teacher, this method **copies the top K=3 layers** from the 12-layer teacher and freezes them, only fine-tuning a tiny (rank $r_a=64$) low-rank adapter.
    * This is a form of *structural pruning* or *layer selection*, not distillation.
    * The 90% parameter reduction (385M $\rightarrow$ 38M) is a direct, trivial consequence of using a K=3 layer model, not a feature of the distillation *method*.
    * The 100x speedup (Table 2) is also misleading. The method is fast because it's only fine-tuning a rank-64 adapter for 3 epochs, whereas the baselines (PKT, RKD, etc.) are training full student models from scratch for many more hours. This is not a fair comparison of distillation *efficiency*; it's a comparison of two completely different training paradigms.

3.  **Over-Engineered and Poorly Justified Methodology:** The paper presents a "kitchen sink" of highly complex techniques without clear justification for each.
    * The **Uncertainty Injection (UI)** mechanism (Section 3.2) is the most glaring example. It involves FFTs, empirical characteristic functions (Eq. 9), an ad-hoc inflation factor (Eq. 11), spectral density inflation (Eq. 12), and multiple projections/decompositions (Eq. 13). The motivation for using frequency-domain ChF differences to model *data subsetting bias* is not explained.
    * The ablation in Table 4 shows that UI provides only a marginal, $\approx$1-2% improvement (e.g., K=12, -0.54% vs -1.17% without UI). This suggests the core results are not dependent on this highly complex component, making the paper feel unnecessarily convoluted.

4.  The paper fails to include the single most important ablation study to validate its core thesis. The method combines (A) layer copying, (B) subspace alignment loss ($L_\mu + L_\Sigma$), and (C) uncertainty injection. The authors claim (B) and (C) are the key contributions. What happens if you *only* do (A)? That is, what is the performance of a baseline that simply copies the top K=3 layers, adds the rank-64 adapter, and fine-tunes it using *only* the standard task loss ($L_{task}$)?
    * It is highly probable that this simple baseline would achieve very similar results, suggesting that the entire complex machinery of consensus subspace projection, Wasserstein loss, and frequency-domain UI provides little to no actual benefit over simple layer-copying and fine-tuning.

5. The core premise of a "consensus subspace" (Fig. 2) is demonstrated on only one model family (MOMENT) using one pre-training objective (MAE). It is a significant leap to claim this is a universal, scale-invariant property of *all* TSFMs.

6.  The student model's ability to outperform the teacher (Table 2) is a common red flag in distillation. It suggests that the "student" (which is just a fine-tuned shallow version of the teacher) is acting as a regularized model on a specific task. This further supports the idea that the method is not a general-purpose distillation but rather a task-specific fine-tuning.

**Questions:**

1.  Can the authors please elaborate on the "bias-resistant" claim? Do they agree with the hypothesis that CSD acts as a strong regularizer by distilling only the low-rank subspace, and is this why the student generalizes better than the (potentially overfit) teacher?

2.  What is the intuition for using the *frequency-domain* ChF difference to model the *data subsetting* bias? Why is this a better model of bias than, for example, a spatial-domain difference? Furthermore, the inflation factor in Eq. 11 seems somewhat ad-hoc. How sensitive is the method to this specific formulation?

3.  How expensive is the offline layer selection process? Have the authors tried a simpler heuristic, such as just selecting the top $K$ layers (e.g., layers $L, L-1, ... L-K+1$), and how does that compare to the marginal contribution method?

4.  To confirm, is the 100x distillation speedup (Table 2) primarily due to 1) requiring fewer training epochs (3 epochs) and 2) eliminating the need for a teacher forward-pass at each training step by using the pre-computed cache $(\mu_T, \Sigma_T)$?

---

> ### Author Response · Authors · 2025-11-20
>
> We sincerely appreciate your thoughtful and detailed evaluation. Thank you for taking the time to review both our manuscript and the accompanying code repository, and for acknowledging the motivation underlying our work.
>
> After carefully considering all reviewers’ constructive feedback, we have decided to substantially revise the manuscript, which we estimate will require approximately one month. Consequently, we will withdraw the current submission from ICLR and resubmit after completing a full revision.
>
> Below we provide brief responses to the key concerns you raised. We hope that, when you see the revised manuscript in the future, these issues will have been addressed in a clearer and more comprehensive manner.
>
> ---
>
> ### Response to Weaknesses
>
> **1. Logo in submitted code.**
> This was an oversight on our part, and the logo has been removed.
>
> **2. Is our method truly a form of distillation?**
> We agree that categorizing the proposed method is challenging. On one hand, we understand the “bait-and-switch” concern you raised. On the other hand, our method explicitly computes the teacher’s consensus subspace and enforces the student adapter’s outputs to align with this subspace. This form of *feature alignment* is mathematically consistent with feature-based distillation.
> In the revision, we will adjust the framing to shift the focus away from strict categorization and toward the substantive insights the method provides for TSFM compression.
>
> **3 & 4. Overly complex methodology and insufficient justification.**
> We will strengthen the explanation and motivation for the more complex components, and design more targeted experiments to isolate and validate the core mechanisms. Additional results and ablations will be included in the appendix.
>
> **5. Generality of the consensus subspace phenomenon.**
> We have conducted similar experiments on additional TSFMs and observed the same trend. We will provide a more thorough discussion of this point in the revised version.
>
> **6. Student outperforming the teacher: red flag or opportunity?**
> We have carefully considered this issue. Interestingly, a recent NeurIPS-accepted work suggests that broader noise in time-series datasets can lead to overfitting in TSFM training [1]. Our method may implicitly act as a regularized alignment mechanism, potentially benefiting from distilled or denoised structure. We will investigate this further and incorporate our findings in the revised manuscript.
>
> ---
>
> ### Response to Questions
>
> In general, we will incorporate your questions into the discussion section of the revised version. For parameters that require sensitivity analysis, we will include more experiments. We will also compare multiple scale-specific heuristics and provide a detailed breakdown of the source of efficiency gains (regarding question 4: reduced FLOPs are the key factor; reduced training epochs are not the main contributor).
>
> ---
>
> In summary, we will approach the revision with a constructive and academically rigorous attitude. We genuinely appreciate the weaknesses and questions you have raised, and we thank you sincerely for your contribution to improving our work. We look forward to the opportunity for further exchange in the future.
>
> **[1]** Fu, Yisong, et al. *"Selective Learning for Deep Time Series Forecasting."* arXiv preprint arXiv:2510.25207 (2025).

---

### Note · Authors · 2025-11-29

I have read and agree with the venue's withdrawal policy on behalf of myself and my co-authors.